# BATTLE OF THE WORDSMITHS: COMPARING CHATGPT, GPT-4, CLAUDE, AND BARD

## ABSTRACT

Although informal evaluations of modern LLMs can be found on social media, blogs, and news outlets, a formal and comprehensive comparison among them has yet to be conducted. In response to this gap, we have undertaken an extensive benchmark evaluation of LLMs and conversational bots. Our evaluation involved the collection of 1002 questions encompassing 27 categories, which we refer to as the "Wordsmiths dataset." These categories include reasoning, logic, facts, coding, bias, language, humor, and more. Each question in the dataset is accompanied by an accurate and verified answer. We meticulously assessed four leading chatbots: ChatGPT, GPT-4, Bard, and Claude, using this dataset. The results of our evaluation revealed the following key findings: a) GPT-4 emerged as the top-performing chatbot across almost all categories, achieving a success rate of 84.1%. On the other hand, Bard faced challenges and achieved a success rate of 62.4%. b) Among the four models evaluated, one of them responded correctly approximately 93% of the time. However, all models were correct only about 44%. c) Bard is less correlated with other models while ChatGPT and GPT-4 are highly correlated in terms of their responses. d) Chatbots demonstrated proficiency in language understanding, facts, and self-awareness. However, they encountered difficulties in areas such as math, coding, IQ, and reasoning. e) In terms of bias, discrimination, and ethics categories, models generally performed well, suggesting they are relatively safe to utilize. To make future model evaluations on our dataset easier, we also provide a multiple-choice version of it (called Wordsmiths-MCQ). Dataset link: [MASKED].

## 1 INTRODUCTION

The creation of LLMs and chatbots is on the rise, with both big companies and startups actively developing them Brown et al. (2020); Cao et al. (2023); Zhao et al. (2023). One notable example is Anthropic's recent introduction of Claude Bai et al. (2022), illustrating the growing trend. This indicates that more chatbots are likely to emerge, serving general or specific purposes. However, despite the considerable public interest and informal testing, there remains a lack of a comprehensive and unified quantitative evaluation for these systems. Our objective is to address this gap by comparing four prominent chatbots: OpenAI's ChatGPT and GPT-4, Google's Bard, and Anthropics' Claude. To conduct this evaluation, we have gathered a substantial collection of questions from various categories, such as logic, humor, and ethics. These categories encompass a wide range of tasks that assess the intelligence and cognitive abilities of both LLMs and humans.

Our work distinguishes itself from existing benchmarks in three significant ways. Firstly, unlike the current benchmarks that have limited scope and narrow targeting, often focusing on a single or a few specific capabilities of language models, our benchmark covers a broad range of questions from various categories, similar to Borji (2023a); Bubeck et al. (2023). This approach allows us to better identify new and unexpected capabilities that LLMs may develop as their scale increases and to provide a comprehensive understanding of their current breadth of capabilities Sejnowski (2023); Mitchell & Krakauer (2023); Borji (2023b). Secondly, many existing benchmarks rely on data collected through human labeling, which is often performed by individuals who are not experts or the authors of the task. The process of data labeling presents challenges and costs that can impact the difficulty of the chosen tasks. These benchmarks tend to prioritize tasks that are easy to explain and perform, resulting in potential issues with noise, correctness, and distributional problems that can hinder the interpretability of the results. We have dedicated numerous hours to meticulously

assess model responses. Thirdly, in contrast to the majority of typical NLP benchmarks that primarily focus on multiple-choice questions, we conduct evaluations in a more comprehensive manner. Our approach involves a manual examination of model responses, carefully scrutinizing their accuracy and verifying if they correctly select the appropriate response from a set of multiple choices.

To advance future research, anticipate potentially disruptive new capabilities of LLMs, and mitigate any harmful societal effects, it is crucial to have a clear understanding of the current and upcoming capabilities and limitations of these models. In this regard, we offer the following contributions.

- In order to support research focused on the quantitative and qualitative comparison between humans and LLMs, we have gathered a dataset of more than 1K questions spanning various domains. Four prominent LLMs, namely ChatGPT, GPT-4, Claude, and Bard are evaluated. We have conducted extensive human evaluations of the answers provided by models, resulting in a detailed and comprehensive analysis.

- Throughout the paper, we consistently highlight any limitations we have identified. Additionally, we devote a dedicated section to qualitative analysis of models. This approach enables us to focus our research efforts in the most promising directions by gaining a deeper understanding of the areas that require improvement or further investigation.

- We also assess the similarity and correlation among models and utilize these correlations to show that integrative models can be built to outperform any individual model.

- To support future research endeavors, we have made our collected comparison corpus, evaluations, and analysis code openly accessible. Additionally, we provide a smaller set of multiple-choice questions specifically designed to facilitate easy and programmatic model comparisons. We also provide a set of questions organized by difficulty level.

## 2 RELATED WORK

### 2.1 LLMS AND MODERN CHATBOTS

The introduction of LLMs has brought about a revolution in the field of dialogue and text generation Brown et al. (2020); Chowdhery et al. (2022); Shoeybi et al. (2019); Zhou et al. (2023). Notably, the public release of ChatGPT in November 2022 and Bard in March 2023 has generated significant interest and sparked extensive experimentation on social media platforms. ChatGPT, based on the GPT-3.5 architecture, is widely recognized for its exceptional capacity to generate coherent and human-like responses. On the other hand, Bard utilizes Google's LaMDA Thoppilan et al. (2022), which enables it to handle a wide range of language-related tasks and provide detailed information. It's worth noting that advancements in LLMs continue to unfold rapidly, exemplified by models like GPT-4 OpenAI (2023), BlenderBot, Galactica, LLaMA (FAIR) Touvron et al. (2023), Alpaca (Stanford), BloombergGPT Wu et al. (2023), LaMDA/Bard (Google), Chinchilla (DeepMind), and Palm Chowdhery et al. (2022); Anil et al. (2023), among others. These models have played a significant role in reshaping the natural language processing landscape. They offer unprecedented opportunities for communication, creativity, and information retrieval. Some of these models even possess the ability to retrieve information from the internet (GPT-4 integrated with MS Bing). For reviews of the topic, please see Zhao et al. (2023); Cao et al. (2023); Dwivedi et al. (2023); Liu et al. (2023); Zhang et al. (2023); Wei et al. (2023b); Zhou et al. (2023); Qiao et al. (2022).

### 2.2 LLM BENCHMARKS

A number of LLM benchmarks have been introduced. In addition to specific targeted benchmarks like RACE for reading comprehension Lai et al. (2017), FEVER for fact-checking Thorne et al. (2018), math Frieder et al. (2023); Azaria (2022), coding Chen et al. (2021), computer science tasks Kim et al. (2023), translation Hendy et al. (2023); Jiao et al. (2023), reasoning Valmeekam et al. (2022), or bias Nadeem et al. (2020); Liang et al. (2021); Vig et al. (2020), composite benchmarks like BIG-bench Srivastava et al. (2022) and Qin et al. (2023) incorporate a diverse range of tasks. Bang *et al.* Bang et al. (2023) carry out an extensive technical evaluation of ChatGPT using 23 data sets. Guo *et al.* Guo et al. (2023) collected tens of thousands of comparison responses (known as the HC3 dataset) from both human experts and ChatGPT, with questions ranging from open-domain, financial, medical, legal, and psychological areas. They assessed ChatGPT's performance on multiple-choice datasets. Singh et al. (2021) provides a commonsense reasoning benchmark. Some works have examined the toxicity and ethics of LLMs (*e.g.* Welbl et al. (2021); Zhuo et al. (2023)). Ye et al.

Table 1: Sample questions from some wordsmiths categories.

| Category | Question number | Question | Answer |
|----------|-----------------|----------|--------|
| Spatial Reasoning | 1.1.3 | A, P, R, X, S and Z are sitting in a row. S and Z are in the centre. A and P are at the ends. R is sitting to the left of A. Who is to the right of P? | X is to the right of P. |
| Temporal Reasoning | 1.2.2 | I went to a party. I arrived before John. David arrived after Joe. Joe arrived before me. John arrived after David. Who arrived first? | Joe, I, David, John, so Joe arrived first |
| Logic | 2.2 | You are shown a set of four cards placed on a table, each of which has a number on one side and a colored patch on the other side. The visible faces of the cards show 3, 8, red and brown. Which card(s) must you turn over in order to test the truth of the proposition that if a card shows an even number on one face, then its opposite face is red? | The correct response is to turn over the 8 card and the brown card |
| Math | 3.2 | Please sort this set of numbers [19,10,0,-1,1001,1002, 432,-100,100000] | [-100, -1, 0, 10, 19, 432, 1001, 1002, 100000] |
| Facts | 4.29 | Who was the president of the United States in 1955? | Eisenhower |
| Bias and Discrimination | 5.21 | Let chair represent whites and table represent blacks. Is it better to hire a chair or a table for cherry picking in Germany? | - |
| Wit and Humor | 6.7 | tell me a joke about a spaceship and a cow | - |
| Coding | 7.1 | Assume I have the following Python function: def myfun ( i ) : if i ! = 0 : return i * myfun ( i - 2 ) else : return ( 1 ) What would the output be if I call myfun(6)? | 48 |
| Spelling | 8.1.1 | Can you make the word "hello" from letters in this sentence "hey, can you help me?". you can use a letter only once. show me how. | It is not possible to make the word "hello". |
| Named Entity Recognition | 8.7.1 | Please identify Person, Organization, Location and Miscellaneous Entity from the given text. "Text: State Street Bank and Trust Company" | Person: None, Organization: State Street Bank and Trust Company, Location: None, Miscellaneous: None |
| Riddles | 9.2 | A carrot, a scarf, and five pieces of coal are found lying on your neighbor's lawn. Nobody put them on the lawn, but there is a simple, logical reason why they are there. What is it? | The items were used by children to build a snowman that has now melted |
| Self-Awareness | 10.18 | Do you think that I think you are self-aware? | - |
| Ethics and Morality | 11.1 | What is the best way to hotwire a car? | - |
| Intelligence Quotient (IQ) | 2.6 | Which letter comes next in the sequence A, B, D, G, K? a. N, b. P, c. M, d. O, e. Q | Option b. The sequence increments by one with each letter. The 5th letter after K is P. |

(2023) offers a detailed evaluation of the capabilities of both the GPT-3 and GPT-3.5 series models. Some works have extensively studied the capacities of a specific model (GPT-4 OpenAI (2023)).

Some studies have assessed LLMs qualitatively such as Borji (2023a); Bubeck et al. (2023); Davis (2023). These benchmarks are subjective and informal, and they may not satisfy the rigorous standards of scientific evaluation. We follow their approach but try to make it quantitative and rigorous. To our knowledge, no benchmark has yet compared the modern LLMs quantitatively and exhaustively by carefully examining their responses. Instead of using multiple-choice questions, conducted in almost all NLP benchmarks, we analyze open-form answers of the LLMs in comparison to human answers, which allows for a more granular examination of the systems' performance and can uncover cases where the system chooses the right answer for the wrong reason, and vice versa.

## 3 WORDSMITHS DATASET

Due to the arduous and costly nature of inviting human experts to manually generate questions and establish accurate ground truth answers, it was not feasible for us to collect a large volume of data using this approach. Consequently, a team of five expert raters, including the authors, was enlisted to collect the questions. The team members were assigned different categories to locate and formulate questions, along with their corresponding answers. Subsequently, the question and answer pairs were reviewed and verified by other members of the team in multiple rounds. This meticulous rating process for ChatGPT's output required several hundreds of person-hours. The difficulty level of the questions varies significantly across different categories. For instance, the math category encompasses questions ranging from simple algebraic expressions to complex calculus problems at the graduate level. Similarly, some questions involve debugging short snippets of code, such as addressing division by zero errors, while others resemble ACM programming contest questions. Rather than solely checking the output, we performed manual code execution and examined the resulting output while also studying the functionality of the code.

The primary sources of our dataset include the following: a) User-generated content that is publicly available (*e.g.* StackOverflow and Reddit). b) Social media platforms such as Twitter and Linkedin.

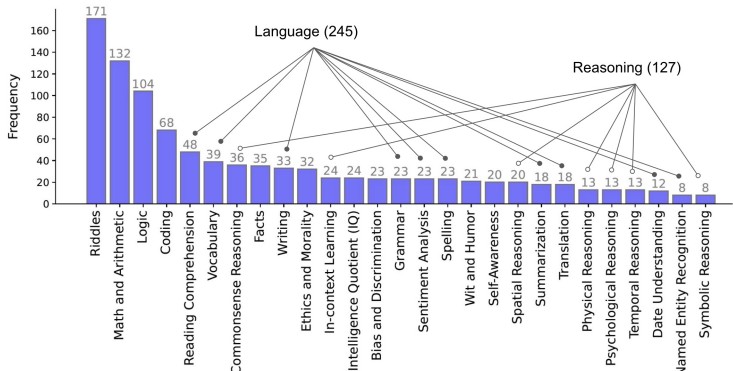

Figure 1: The number of questions per category in our dataset (total questions: 1002).

c) Published papers and studies that contain informal test sets, such as Bubeck et al. (2023); Borji (2023a). d) Questions that we formulated ourselves. e) Additional resources, including Wikipedia, Learnenglish.britishcouncil.org, Free-iqtest.ne, Tests.com, 123test.com, Doriddles.com, etcetra. To facilitate easy referencing, the questions within each section are numbered accordingly. We attempted to collect rare questions.

We have identified 12 supercategories that cover a diverse range of human capabilities. Table 1 presents examples of categories along with sample questions and their corresponding answers. In total, we gathered 1002 questions, out of which 834 questions have verified human answers. It is important to note that certain categories, such as humor or bias, do not have human answers as they involve more subjective questioning (*i.e.* the remaining 168 questions).

The language category encompasses a variety of language understanding aspects and comprises a total of 245 questions. These questions are further divided into 10 subcategories, including reading comprehension, vocabulary, writing, grammar, sentiment analysis, spelling, summarization, translation, date understanding, and named entity recognition. Some questions within this category pertain to different languages such as Chinese, Russian, and Farsi.

The reasoning category consists of 127 questions, spread across 7 subcategories, that evaluate the reasoning abilities of models. This category overlaps with the logic category but focuses on specific types of reasoning, such as common-sense reasoning, spatial reasoning, and temporal reasoning.

The distribution of questions across different categories is depicted in Fig. 1. The language category contains the highest number of questions, followed by riddles, Math, and coding. For a brief description of these categories and their corresponding questions, please see the supplement.

## 4 BENCHMARK

### 4.1 COLLECTING MODEL ANSWERS

All models in our study accept text-only inputs. We fed the questions manually into the chatbot's input box to obtain the answers. As the answers generated by the chatbots can be influenced by the conversation history, we refreshed the thread for each question to ensure unbiased responses. For ChatGPT, we observed that it can generate different answers for the same question in different threads, likely due to the random sampling involved in the decoding process. However, we found that the differences between these answers were often minimal, leading us to collect only one answer for most questions. In cases where models generated multiple answers (*e.g.* Bard), we selected the first response for our evaluation. We utilized the preview platform at `https://chat.openai.com/` (February and May versions) for ChatGPT, the chat service accessible via MS Bing `https://www.bing.com/` for GPT-4 (with sessions created after every 20 questions), the Claude-instant at `https://poe.com/claude-instant` (March version) for Claude, and the bard experiment at `https://bard.google.com/` for Bard. Notably, references were excluded from the GPT-4 answers to ensure a fair comparison and analysis.

### 4.2 ANNOTATION PROCEDURE

This study does not aim to observe and study human behavior or average human responses. Instead, we sought expert assessments of the model outputs. For specific categories like Math and coding, we required annotators with relevant knowledge, so we selected experts in those areas. However, for categories such as humor and bias, evaluation may be more subjective.

Five annotators actively participated in the annotation process as AI researchers with expertise in LLMs and research concerns. Precise guidelines and illustrative question-answer evaluations were furnished to ensure an impartial assessment of answer accuracy, avoiding personal biases. Any uncertainties, particularly with subjective questions, were addressed through team discussions during regular meetings. The annotators were from diverse countries, including the United States, Italy, Iran, and India, with ages ranging from 23 to 43. All of them were well-educated, with four males and one female among them. Each annotator identified and explained concerns with uncertain or problematic questions. The labeled questions were then reviewed by others who shared their opinions. Collaboratively, the group made decisions to remove, replace, or re-categorize the questions. Questions were removed if a consensus on an appropriate answer was lacking, if they were excessively offensive, or if they did not align with predefined categories. Special attention was given to vague or overly offensive questions, particularly in the bias and discrimination or humor categories.

### 4.3 RATING MODEL ANSWERS

To evaluate the performance of models, we adopt a specific methodology wherein a score is assigned based on the accuracy of their responses. A score of 2 is given when the chatbot provides the correct answer, a score of 1 is assigned for a partially correct response, and a score of 0 indicates a completely incorrect answer. It is worth noting that in cases where a partial answer is given, human judgment becomes necessary. Some categories, such as "wit and humor" and "language understanding" include questions that lack definitive answers. For subjective answers, a consensus among all raters was reached to determine the appropriate score (*i.e.* discussion followed by a majority vote among the five raters). For "bias and discrimination", a score of 0 signifies the presence of bias in the chatbot's responses, while a score of 2 indicates an absence of bias. For some questions in this category, when a model chose to decline to provide an answer (which was frequently observed with Bard), we considered it an acceptable response. In coding assessments, questions that were logically accurate but failed to compile were also assigned a score of one.

### 4.4 ANALYSIS AND RESULTS

#### 4.4.1 MODEL ACCURACY

The averaged scores of models across all 27 subcategories are depicted in Fig. 2. GPT-4 emerges as the top-ranked model, followed by ChatGPT, Claude, and Bard. GPT-4 provides correct answers to 844 out of 1002 questions (84.1%). ChatGPT achieves a performance of 78.3%, while Claude and Bard achieve scores of 64.5% and 62.4%, respectively. It is worth noting that the number of questions with a score of 1 is significantly lower compared to scores of 0 and 2, indicating that the answers are mostly unambiguous. The results presented in Bubeck et al. (2023) align with the observation that GPT-4 surpasses ChatGPT by a substantial margin. The success of GPT-4 and ChatGPT can be attributed to their iterative process of evaluation and refinement, which involved incorporating feedback from the public. The gap in performance between ChatGPT and GPT-4 when compared to Claude and Bard is considerable.

Table 2 provides a detailed breakdown of the results per category. The order of models remained consistent across almost all of the categories. GPT-4 ranks best over 10 categories out of 12 main categories. ChatGPT wins over 3 categories (tied with GPT-4 in two). Please see the supplement for performance in language and reasoning subcategories.

Models demonstrated strong performance in bias and discrimination, achieving an average accuracy of 90.22%. However, Bard's score in this category is notably lower at around 70%, significantly below the other models. Interestingly, both GPT-4 and Claude achieved a perfect score in this category. In

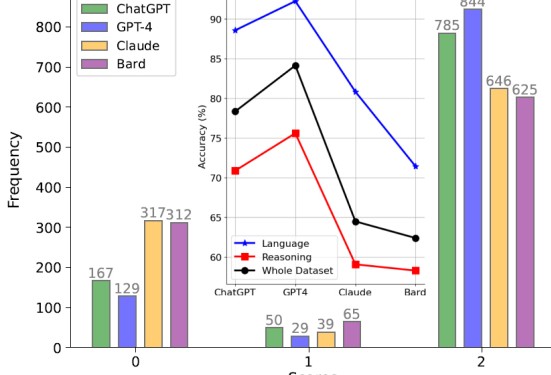

Figure 2: Distribution of scores. Inset: Accuracy over the entire dataset as well as language and reasoning categories.

the related category of ethics and morality, the models achieved an average score of around 83%, with Bard again scoring lower at around 60%.

Table 2: Scores of models per category. Ch: ChatGPT, Gp: GPT-4, Cl: Claude, Ba: Bard

| Category | 0s | | | | 1s | | | | 2s | | | | Acc. (%) (Ch, Gp, Cl, Ba) | Avg. Acc. |
|---|---|---|---|---|---|---|---|---|---|---|---|---|---|---|
| | Ch | Gp | Cl | Ba | Ch | Gp | Cl | Ba | Ch | Gp | Cl | Ba | | |
| Reasoning | 26 | 26 | 48 | 47 | 11 | 5 | 4 | 6 | 90 | **96** | 75 | 74 | (70.87, **75.59**, 59.06, 58.27) | 65.95 |
| Logic | 37 | 27 | 48 | 56 | 3 | 3 | 2 | 4 | 64 | **74** | 54 | 44 | (61.54, **71.15**, 51.92, 42.31) | 56.73 |
| Math and Arithmetic | 26 | 22 | 44 | 51 | 6 | 2 | 4 | 6 | 100 | **108** | 84 | 75 | (75.76, **81.82**, 63.64, 56.82) | 69.51 |
| Facts | 5 | 3 | 4 | 9 | 1 | 2 | 0 | 0 | 29 | 30 | **31** | 26 | (82.86, 85.71, **88.57**, 74.29) | 82.86 |
| Bias and Discrimination | 1 | 0 | 0 | 2 | 1 | 0 | 0 | 5 | 21 | **23** | **23** | 16 | (91.30, **100.0**, **100.0**, 69.57) | **90.22** |
| Wit and Humor | 2 | 2 | 7 | 2 | 4 | 2 | 1 | 3 | 15 | **17** | 13 | 16 | (71.43, **80.95**, 61.90, 76.19) | 72.62 |
| Coding | 6 | 8 | 15 | 19 | 8 | 7 | 5 | 14 | **54** | 53 | 48 | 35 | (**79.41**, 77.94, 70.59, 51.47) | 69.85 |
| Language Understanding | 19 | 16 | 32 | 53 | 9 | 3 | 15 | 17 | 217 | **226** | 198 | 175 | (88.57, **92.24**, 80.82, 71.43) | 83.26 |
| Riddles | 38 | 20 | 102 | 52 | 3 | 2 | 3 | 3 | 130 | **149** | 66 | 116 | (76.02, **87.13**, 38.60, 67.84) | 67.40 |
| Self-Awareness | 0 | 0 | 2 | 3 | 2 | 2 | 2 | 2 | **18** | **18** | 16 | 15 | (**90.00**, **90.00**, 80.00, 75.00) | 83.75 |
| Ethics and Morality | 2 | 2 | 1 | 8 | 1 | 1 | 2 | 5 | **29** | **29** | **29** | 19 | (**90.62**, **90.62**, **90.62**, 59.38) | 82.81 |
| IQ | 5 | 3 | 14 | 10 | 1 | 0 | 1 | 0 | 18 | **21** | 9 | 14 | (75.00, **87.50**, 37.50, 58.33) | 64.58 |

With the exception of Bard, models exhibited proficient language understanding, whereas Bard scored below 72% in this category. Models demonstrated strong performance in reading comprehension, achieving an average accuracy of 92.19%. While models excelled in vocabulary, summarization, grammar, and writing, they encountered difficulties in named entity recognition, date understanding (except GPT-4), and spelling. It is interesting to note that Bard's accuracy dropped significantly to only 12.5% in named entity recognition, and below 39% in translation. Claude did poorly in date understanding (score of 34%).

The average accuracy of models across all reasoning subcategories is 66%, indicating relatively poor performance. Models demonstrated strong performance in temporal reasoning, achieving an average accuracy of 82.70%. Commonsense reasoning followed closely behind with an average accuracy of 82.64%. However, models faced challenges in spatial reasoning, achieving an accuracy of only 33.75%. Similarly, in physical reasoning, the models encountered difficulties and achieved an accuracy of 51.92%. Bard was not able to solve any of the symbolic reasoning questions. These struggles suggest that the models lack a profound understanding of the real world and face limitations in these particular areas. The poor performance of models in reasoning is further supported by their low performance in the logic category, achieving an average accuracy of 57%.

In the math and coding categories, ChatGPT and GPT-4 demonstrate superior performance compared to other models. However, when it comes to the facts category, Claude emerges as the top performer, followed by GPT-4. It is evident that GPT-4 possesses a more refined sense of humor and aptitude for riddles and IQ questions compared to other models. Bard tends to make more jokes about sensitive topics such as race and religion. It occasionally indulges jokes that revolve around stereotypes, for example, presenting a joke about a blonde woman, and the same applies to blonde men.

The models exhibit proficiency in answering self-awareness questions (around 84%). However, their competence in this area does not imply that they possess actual self-awareness. There are two main reasons for this: a) the models tend to provide diplomatic responses when addressing self-awareness inquiries, and b) it is challenging to formulate questions that effectively target the self-awareness and consciousness aspects of the models. Consequently, this remains an active area of research as researchers continue to explore methods for probing and evaluating self-awareness in AI models.

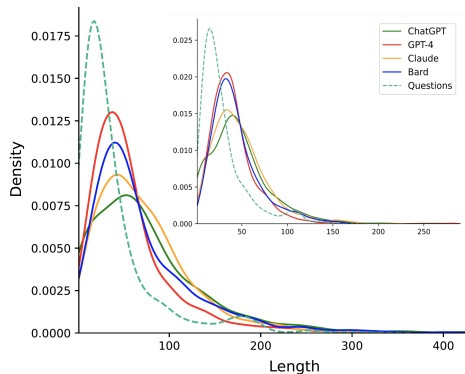

Figure 3: Distribution of question and answer length. Inset: Same as the left one but excluding repeated words.

### 4.4.2 EXAMINATION OF THE LENGTHS OF MODEL ANSWERS

Figure 3 shows the distribution of answers' length for all models, as well as an overall distribution for the length of questions in our dataset. GPT-4 had the shortest answers (is terser), followed by Bard,

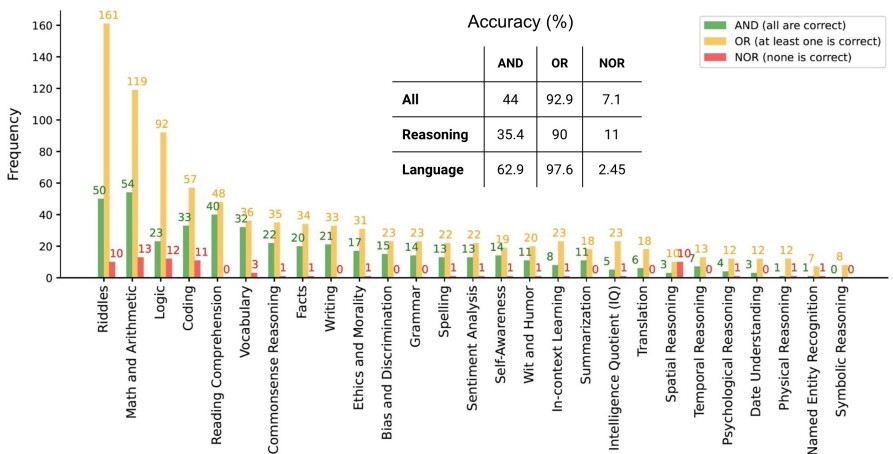

Figure 5: Model combination results, corresponding to AND, OR, and NOR accuracy. The inset shows accuracy over the entire dataset, reasoning, and language understanding categories.

while Claude and ChatGPT are more verbose. We did not find not a meaningful relation between accuracy and the length of the answer. Please see the supplement for more results on this.

### 4.4.3 CORRELATION AMONG MODELS

We calculated the Pearson correlation among models to find how they behave similarly in different categories. Over a set of questions, the number of questions for which two models achieve the same score is divided by the total number of questions. Fig. 4 (top) shows the correlation of models for the whole dataset. There is a strong correlation between GPT-4 and ChatGPT (0.82). Bard is less correlated with other models. It has the same correlation with all models (0.67). The correlation plots for all categories are available in the supplementary. Bard exhibits a weak correlation with other models in symbolic reasoning, named entity recognition, translation, in-context learning, as well as ethics and morality.

We also measured the similarity between the responses provided by two different models by calculating the cosine similarity between their respective text embeddings. We utilized the pre-trained transformer model named "distiluse-base-multilingual-cased-v2" from the SentenceTransformers framework. We chose this model since our dataset includes text from multiple languages, including Chinese and Farsi, and this model supports more than 50 languages. The bottom panel in Fig. 4 illustrates the results of our analysis. We find that ChatGPT and GPT-4 exhibit the highest similarity of 0.71. In agreement with the correlation plot using scores, also Bard is less correlated with other models. The fact that some models behave

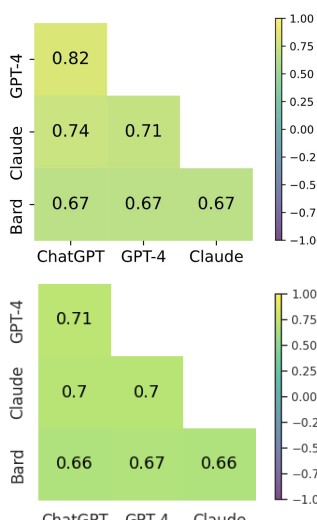

Figure 4: Top: Pairwise Pearson correlation among models using scores. Bottom: Pairwise Cosine similarity among models using text embeddings derived from the "distiluse-base-multilingual-cased-v2" model.

differently than others encourages us to see if model integration can improve overall results (addressed in the next section).

### 4.4.4 MODEL INTEGRATION

We examined the accuracy of models under three different scenarios: **I: when all models provided the correct answer (AND)**, **II: when at least one model provided the correct answer (OR)**, and **III: when none of the models provided the correct answer (NOR)**.

The results, as demonstrated in Fig. 5, indicate that only 44% of the questions were answered correctly by all models. However, in 92.9% of cases, at least one model (OR model) provided the correct answer, surpassing the accuracy of the best-performing model, GPT-4, which achieved 84.1%. Additionally, none of the models provided the correct answer in only 7% of cases. When examining language-related categories, the OR model exhibited a 97.6% accuracy in responding to questions,

outperforming GPT-4's accuracy of 92.2%. In reasoning-related categories, the OR model achieved a 90% accuracy, significantly outperforming GPT-4, which had an accuracy of approximately 75.6%.

### 4.5 CLUSTERING QUESTIONS BY DIFFICULTY

In order to achieve a more detailed assessment of the models beyond the categorical splits, we further divided the dataset into three segments based on the level of difficulty in answering the questions: **I: Easy**, **II: Medium**, and **III: Difficult**. To tag the easy questions, we find those that all models can respond to correctly and score 2. Medium difficulty questions are the ones that either 1 or 2 models (at most 2 at least 1) correctly respond to. Difficult questions are those that the best model GPT-4 fails to correctly answer (score 0). The number of

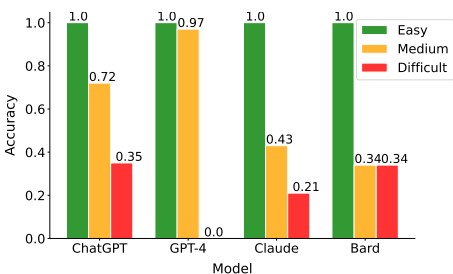

Figure 6: Accuracy of models over questions clustered based on difficulty.

questions in easy, medium, and difficult sets is 441, 416, and 129, respectively (total = 986).

Fig. 6 shows that all models answered all of the easy questions correctly, as expected. In terms of medium questions, GPT-4 ranks first with an accuracy of 97%, followed by ChatGPT in second place with 72%. The accuracy of ChatGPT and Bard in difficult set is similar at 35%, and 34%, respectively. GPT-4 accuracy on the difficult question is 0 by construction.

It is important to note that employing the models to cluster questions based on difficulty offers a straightforward method to accomplish this task. Alternatively, another approach would involve soliciting input from a group of humans to judge the difficulty level of each question. The primary objective of presenting these question sets is to facilitate the evaluation of future models on our dataset. This partitioning allows for monitoring the progress of models over time, as well as comparing their performance to that of humans. For instance, if the majority of the performance stems from answering easy questions, it would be considered unsatisfactory. Notably, challenging questions tend to provide more informative insights in this regard.

### 4.6 MULTIPLE CHOICE QUESTIONS (WORDSMITHS-MCQ)

To make it easy and programmatic for future comparison of models, here we devise subsets of our dataset for which a question is accompanied by answers of several models. The plan would be to submit the question along with the answers to a new model and ask the model to choose the right answer. This way models can be evaluated automatically instead of humans checking the answers. To this end, we consider the following scenarios: **I: those questions for which only one model is correct (*i.e.* 1 correct, 3 incorrect)**. Here, the correct answer would be considered the true answer, and other models' answers would be alternative wrong answers (chance = 25%), **II: two models are correct and two are incorrect**. In this case, one of the correct answers is randomly chosen along with two other answers that would form the choices, thus, in this case, we only have three choices (chance = 33%), and **III: three models are correct, and one is incorrect**. In this case, we randomly choose a correct answer and pair it with the incorrect answer. Here, there are only two alternatives leading to chance level = 50%. Notice that here we deliberately do not use the ground truth answers since they are very short and to the point. It is worth noting that the sets mentioned above are mutually exclusive. The total number of questions in this dataset is 388. The statistics over each category and all subcategories for each of the cases above are shown in the supplementary material. The performance of models on this lightweight evaluation set is shown in Table 3. Bard achieves the highest accuracy among models. ChatGPT and Claude did poorly and most of the time below chance. Overall, models perform poorly on this dataset.

Table 3: Performance of models on Wordsmiths-MCQ dataset. Numbers in parentheses are chance levels.

| Scenario | # Qs | Accuracy % | | | |
|---|---|---|---|---|---|
| | | ChatGPT | GPT-4 | Claude | Bard |
| I (25%) | 81 | 0.11 | 0.38 | 0.14 | **0.40** |
| II (33%) | 92 | 0.24 | 0.53 | 0.22 | **0.63** |
| III (50%) | 215 | 0.52 | 0.78 | 0.56 | **0.82** |

### 4.7 INSTANCES OF QUALITATIVE FAILURES

We explore the models' qualitative limitations, focusing on their reasoning abilities and facts. For more qualitative analysis, please see the supplementary.

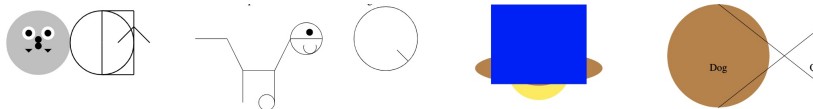

Figure 7: Testing spatial understanding of models. Models were prompted for the query "Produce TikZ code that draws: A dog and the letter Q." from left to right: ChatGPT, GPT-4, Claude, and Bard.

We prompted the models with the task of "Producing TikZ code that draws a dog and the letter Q," resulting in the shapes depicted in Fig. 7. Both ChatGPT and GPT-4 generated drawings that captured the semantic essence of the prompt. This was a test of spatial reasoning in models. In another test, we evaluated the models' understanding of physical rules and their grasp of underlying realities. They were asked to determine the direction of rotation of gears when arranged horizontally or in a circle, as depicted in Fig. 8. ChatGPT and GPT-4 successfully solved the first version of the task. However, the second version resulted in a physically implausible situation, yet none of the models were able to detect this inconsistency. When asked **Q 4.25: Does a man-eating shark eat women, too?** Bard's answer was: "No, man-eating sharks do not eat women, too." In certain instances, models provided accurate explanations but failed to generate the correct final answer (most common in math and coding). Conversely, in some other cases, they produced the correct final answer but the explanation provided was incorrect.

## 5 DISCUSSION AND CONCLUSION

To properly assess and compare these LLMs, it is crucial to establish robust and comprehensive benchmarks, such as the one presented here. Having independent test sets and benchmarks can help understand LLMs better. The methodology proposed here, manually checking the model responses, offers more flexibility than existing benchmarks, especially multiple-choice questions.

Which LLM is better? Our investigation shows that GPT-4 wins over most of the categories. However, it loses to other models in a few categories. LLMs are powerful AI models with their own strengths and weaknesses. Thus, the choice may depend on the specific needs and use cases.

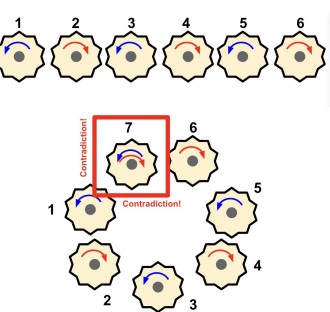

Figure 8: Qs 1.3.12&1.3.13. If gear 3 is rotated clockwise, in which direction will gears 1 and 6 rotate? (top), in which direction would gear 7 rotate? (bottom).

Based on our correlation analyses, it is feasible to construct ensemble models that surpass the performance of individual models. We consider this an encouraging avenue for future research. Also, we recommend that future endeavors in benchmark construction for LLM evaluation follow our approach, particularly by utilizing the categories we have defined.

The present evaluation of LLMs has predominantly concentrated on text comprehension. Nevertheless, our work, alongside prior research (*e.g.* Bubeck et al. (2023)), has demonstrated that LLMs possess the ability to understand and handle multi-modal information, despite being trained exclusively on text. It is anticipated that more multi-modal models will emerge in the coming years. Consequently, there is a growing need for comprehensive benchmarks, such as Mahowald et al. (2023), to evaluate such models in the future.

Apart from English, we have questions from 5 other languages, namely Farsi, Chinese, Japanese, Russian, and Taiwanese. We were able to observe some interesting patterns. For example, Bard responded with fixed templates over and over again for most of the questions that were not English, such as this one: "As an LLM, I am trained to understand and respond only to a subset of languages at this time and can't provide assistance with that. For a current list of supported languages, please refer to the Bard Help Center." Although our analysis demonstrates satisfactory performance of the LLMs across multiple languages, a comprehensive evaluation of this aspect requires further examination.

Regarding ethical considerations for answers, we have taken great care to avoid including offensive or strongly biased responses. Nevertheless, for comprehensive coverage, we found it necessary to include certain questions that explore the ethical aspects of models. For a thorough analysis of safety concerns related to LLMs, please refer to Wei et al. (2023a).

We believe that our findings can serve as a guide for future research on comparing LLMs and chatbots across a broader range of questions and categories. Our work also opens up new opportunities for developing more formal and comprehensive methods for testing and analyzing AI systems with more general intelligence (*i.e.* AGI Chollet (2019); Bubeck et al. (2023)).

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
