# OpenReview forum: "Battle of the Wordsmiths: Comparing ChatGPT, GPT-4, Claude, and Bard"
_ICLR.cc/2024/Conference — ICLR 2024 Conference Withdrawn Submission_

### Official Review · Reviewer_FZSs · 2023-11-01

**Soundness:** 3 good
**Presentation:** 3 good
**Contribution:** 2 fair
**Rating:** 5
**Confidence:** 3

**Summary:**

This paper provides a benchmark for evaluating LLM. The benchmark is claimed to be hand-crafted and optimized for LLM.

**Strengths:**

Unified and standardized benchmarks for LLM are important today and the paper made an effort to contribute in this area. The evaluation in appendix looks very extensive.

**Weaknesses:**

My primary concern is the technical depth and the entire approach smells quite ad-hoc; evaluating the competence/intelligence of a certain subject is a centuries-old area done extensively in education school. Technically, there is not really much machine learning

It would be hard to imagine this level of technical depth could stand any chance for premier ML conferences just a few years back. But I acknowledge that the authors could need to tackle substantial engineering challenges that cannot be easily seen by reviewers.

**Questions:**

I dont know what questions to ask. Philosophically, how is the principle used to build benchmark here different from the benchmark used to build SAT tests?

---

### Official Review · Reviewer_Kq4d · 2023-11-04

**Soundness:** 2 fair
**Presentation:** 3 good
**Contribution:** 2 fair
**Rating:** 5
**Confidence:** 4

**Summary:**

The paper aims to conduct a comprehensive evaluation of the most capable LLMs. It collected 1002 questions encompassing 27 categories include reasoning, logic, facts, coding, bias, language, humor, and more. The dataset is named “Wordsmiths dataset.” Each question in the dataset is accompanied by an accurate and verified answer. The paper evaluated four leading chatbots: ChatGPT, GPT-4, Bard, and Claude. It has several key findings: a) GPT-4 is the top-performing one across all categories, achieving a success rate of 84.1% whereas Bard obtained only 62.4%. b) Among the four models evaluated, at least one of them responded correctly 93% of the time. However, all models were correct only about 44%. c) Bard is less correlated with other models while ChatGPT and GPT-4 are highly correlated in terms of their responses. d) Chatbots demonstrated proficiency in language understanding, facts, and self awareness. However, they did perform well in areas such as math, coding, IQ, and reasoning. e) About bias, discrimination, and ethics categories, models generally did well.

**Strengths:**

1. The paper conducted a comprehensive evaluation of the leading four LLMs, ChatGPT, GPT-4, Bard, and Claude encompassing reasoning, logic, facts, coding, bias, humor, etc.

2. The paper presents several findings on the current capabilities of these models, and their relative performance. Interestingly, there is opportunities to ensemble them: at least one of them responded correctly 93% of the time whereas all models were correct only about 44%.

**Weaknesses:**

1. The paper is very dismissive of the current benchmarks such as HELM, Big-Bench without further elaboration. These benchmarks cover many of the same categories, reasoning, logic, facts, coding, etc. Crucially, how do the results from this paper compare with those from HELM, Big-Bench? Are there any surprises that are not known from existing benchmarks? I would suggest adding a section in the main text or appendix to discuss this.

2. The prompts and annotations are done by a small number of experts including the authors. Could there be significant biases? It seems that samples without consensus are removed. This itself could lead to biases.

3. The paper did not seem to discuss prompt engineering for different LLMs. Specifically, Claude is known to be very strict on the format.
Claude is trained to fill in text for the Assistant role as part of an ongoing dialogue between a human user (Human:) and an AI assistant (Assistant:). Without this structure, Claude doesn't know what to do or when to stop, so it just keeps on going with the arc that's already present.

**Questions:**

On section 6.1, multiple choice questions, Bard achieves the highest accuracy among models. ChatGPT and Claude did poorly and most of the time below chance. This seems to contradict the earlier evaluation that GPT4 is the most capable model. Can you dig deep and resolve the riddle?

Claude-instant is a weaker model compared with Claude. Can you evaluate Claude-2 instead?

"Precise guidelines and illustrative question-answer evaluations were furnished to ensure an impartial assessment of answer accuracy, avoiding personal biases."  Please provide the details guidelines.

"The labeled questions were then reviewed by others who shared their opinions. Collaboratively, the group made decisions to remove, replace, or re-categorize the questions." What criteria are used? Please share those that are removed and replaced.

---

### Official Review · Reviewer_NUi4 · 2023-11-05

**Soundness:** 2 fair
**Presentation:** 3 good
**Contribution:** 2 fair
**Rating:** 3
**Confidence:** 3

**Summary:**

This paper introduces Wordsmiths, a benchmark for evaluating four leading chatbots -- ChatGPT, GPT-4, Claude, and Bard. This dataset contains 1002 questions across 27 diverse categories, including reasoning, logic, facts, coding, bias, language, humor, etc. The paper highlights the strengths and weaknesses of each model across these categories and assesses their correlation. The authors have also created the Wordsmiths-MCQ, a multiple-choice version of the dataset to facilitate future model evaluations. Their evaluation helps identify the current capabilities and limitations of these models, providing valuable insights and directions for future research.

**Strengths:**

1. The proposed benchmark could potentially be useful to evaluate the capabilities of large language models.
2. The paper is well-structured and clearly written. The methodology and evaluation process are well-explained.

**Weaknesses:**

1. The paper only compares four language models, there are other public models available that could have also been included in the study, allowing for a more comprehensive evaluation of the current state-of-the-art landscape.
2. The authors manually evaluate the model responses, which brings inherent subjectivity to the evaluation process. The paper could benefit from including clearer guidelines on how disagreements among evaluators were resolved and considerations made to maintain a consistent evaluation. Even though multiple-choice option was presented, many studies have shown that multiple-choice evaluation is brittle.
3. The questions are mainly from online sources, which could already be included in the training data. Therefore, data contamination is a cofounder.

**Questions:**

1. How were disagreements among evaluators resolved during the manual evaluation process? Were there any specific guidelines or measures taken to ensure consistency and mitigate subjectivity in the evaluation? Would be nice to share the guidelines.

---

### Official Review · Reviewer_NvBJ · 2023-11-09

**Soundness:** 2 fair
**Presentation:** 3 good
**Contribution:** 3 good
**Rating:** 3
**Confidence:** 3

**Summary:**

This paper proposes the Wordsmiths dataset, a collection of 1002 questions covering 27 categories. Several popular chatbot applications, including ChatGPT, GPT-4, Claude, and Bard, were empirically compared using this dataset.

**Strengths:**

The key contribution is the dataset, particularly the fact that the answers are examined by humans. It can serve not only as a benchmark for generative AI models but can also be used to fine-tune these models.

**Weaknesses:**

- Some details of the dataset, including those concerning the annotation process, are missing.
- The comparison conducted on popular chatbot applications leaves room for improvement.

**Questions:**

- The description in the text suggests the proposed dataset is multilingual. What is the language distribution of the questions and answers?
- The annotation of the dataset involves multiple annotators. While the paper describes the general manual annotation procedure, it provides few statistics in this regard. For example, what is the inter-annotator agreement? (Some questions in the dataset are subjective and rather open-ended. How is inter-annotator agreement even measured in such cases?) And more generally, are there quantitative guidelines to resolve disagreements and reach consensus?
- While the comparisons between popular chatbot applications reveal interesting insights, reproducibility is a key issue. The paper provided the rough time (e.g., in which month) when the chatbots were evaluated, but without knowing the versions of the underlying models, it is challenging, if not impossible, for the reader to reproduce the results. The fact that all these chatbots rapidly iterate their underlying models only exacerbates this issue.
- Figure 6: While being the strongest model in general, why does GPT-4 have a 0 score for difficult questions?
- Table 3: The results on the multiple-choice data seem rather implausible. Given that these are essentially the same questions, why do the accuracies here show a completely different picture? Please elaborate.